# Caregiver Knowledge, Attitude, and Behavior toward Care of Children with Cerebral Palsy: A Saudi Arabian Perspective

**DOI:** 10.3390/healthcare12100982

**Published:** 2024-05-10

**Authors:** Abdulaziz Almosallam, Ahmad Zaheer Qureshi, Bashayer Alzahrani, Sultanh AlSultan, Waad Ibrahim Alzubaidi, Alanoud Alsanad

**Affiliations:** 1Department of Internal Medicine, Majmaah University, Majmaah 11952, Saudi Arabia; 2Department of Physical Medicine and Rehabilitation, King Fahad Medical City, Riyadh 11525, Saudi Arabia; qureshipmr@gmail.com; 3Department of Mental Health, Sultan Bin Abdulaziz Humanitarian City, Riyadh 13571, Saudi Arabia; baalzahrani96@gmail.com; 4Department of Patient Experience, Sultan Bin Abdulaziz Humanitarian City, Riyadh 13571, Saudi Arabia; sultanah1984@gmail.com; 5Department of Research and Scientific Center, Sultan Bin Abdulaziz Humanitarian City, Riyadh 13571, Saudi Arabia; waadalzubaidi1@gmail.com; 6Department of Patient Affairs, Sultan Bin Abdulaziz Humanitarian City, Riyadh 13571, Saudi Arabia; anoudalsanad@gmail.com

**Keywords:** cerebral palsy, caregivers, knowledge, attitude, behavior, Saudi Arabia

## Abstract

The care of children with cerebral palsy (CP) requires a complex system of care that is not only dependent on health care resources, but is also strongly influenced by social and cultural attributes. Hence, it is important to explore the understanding and practices of caregivers within a regional perspective. This study was conducted to investigate the knowledge, attitude, and behavior (KAB) of parents with children diagnosed with CP in Saudi Arabia. A cross sectional survey was conducted on the caregivers of children with CP admitted for inpatient rehabilitation between October 2023 to January 2024. A total of 216 caregivers participated in this survey. About 82.9% of caregivers were the mothers of CP children, half (50.5%) were ≤36 years old, 53.7% were highly educated, and 89.2% lived in urban areas. More than half of the participants (57.7%) owned their homes. Regarding children, spastic quadriplegia was the most common type (46.3% of cases). Overall, the participants recorded good values for all variables for KAB. The mean value for attitude was higher (2.67 ± 0.20) when compared to behavior (2.49 ± 0.36) and knowledge (2.46 ± 0.25). Participants who had children with spastic quadriplegia CP reported lower behavior scores than their peers. Strategies with a special emphasis on improving the behaviors of caregivers for children with quadriplegia need to be adapted. Similarly, the living situations of families need to be taken into consideration given its significant association with the attitude of caregivers. A considerable lack of knowledge in handling emergency situations by caregivers signifies a gap in care, which could have potentially life-threatening consequences.

## 1. Introduction

Cerebral palsy (CP) is a permanent and non-progressive disorder that occurs within the developing fetal or infant brain. It is one of the most common causes of physical disability, and it is often associated with motor disorders, which are accompanied by disturbances to sensation, perception, cognition, and communication [1]. Depending upon the severity of impairments, children may have complications like spasticity, contractures, scoliosis, joint dislocations, skin breakdown, aspiration pneumonia, malnutrition, dental problems, low bone mass, epilepsy, psychological problems, and bowel and bladder complications [2]. This, in turn, affects their activities of daily living and mobility, and it can limit their social integration. Not only is the patient is affected, but there is also a considerable impact on families and caregivers [3,4,5,6]. The effects are reciprocal. For children with CP who are dependent on others, the awareness, knowledge, and behavior of caregivers can profoundly impact their care. This emphasizes the need for a family-centered approach. It is one of the most important strategies in pediatric rehabilitation as it actively acknowledges and addresses family member concerns in a holistic manner [3].

Previous studies have shown that the severity of disability influences the needs of the families in looking after their children with CP [7,8,9,10,11,12]. The level of the child’s functional ability is usually classified using the Gross Motor Functional Classification System (GMFCS) [13]. When compared to GMFCS Levels I, II and III, children with GMFCS Levels IV and V are highly dependent and require additional health care resources and higher medication cost [14]. The management of CP often requires specialized medical, rehabilitative, social, and educational services in order to deliver better outcomes [15]. The support and involvement of families and the surrounding community are crucial for their long-term well-being and integration into society [16,17]. Families of children with CP, particularly the parents who serve as caregivers, play a pivotal role in facilitating the developmental milestones of children with CP. Socioeconomic status, marital discord, and other parental responsibilities can impact the care of children with CP. Furthermore, the level of understanding of parents about CP affects the care continuum [18,19]. A thorough understanding of CP and its associated impairments can foster positive attitudes and exhibit constructive behaviors [20].

The prevalence of CP in Saudi Arabia has been estimated to range from 1.6–4.3 per thousand live births [21]. Various international studies have explored the knowledge, attitude, and behavior (KAB) of caregivers toward the care of children with CP [22,23,24,25,26,27,28]; however, the local data in this regard are lacking [29,30,31]. The care of children with CP requires a complex system of care that is not only dependent on health care resources, but is also strongly influenced by social and cultural attributes. Hence, it is important to explore the understanding and practices of caregivers within a regional perspective. Since Saudi Arabia shares similar social, cultural, and religious practices attributes with many neighboring countries in the Middle East, the findings of this study can also help to optimize the care of CP children in this part of the world.

The objective of this study was to investigate the knowledge, attitude, and behavior of parents with children diagnosed with CP in Saudi Arabia.

## 2. Materials and Methods

### 2.1. Type

This is a cross sectional hospital-based study that was conducted in accordance with patient acceptance.

### 2.2. Setting and Participants

The caregivers’ knowledge, attitude, and behavior were assessed using a questionnaire that was developed and validated at Sultan Bin Abdulaziz Humanitarian City (SBHC). Children aged 1–14 years with a confirmed diagnosis of CP according to the International Classification of Diseases (ICD-10) were included in the study. Children that had both CP and other neurological disorders were excluded. The survey was conducted with the caregivers of children who fulfilled the inclusion criteria and were admitted to SBAHC between October 2023 to January 2024. SBAHC is the largest rehabilitation facility (>300 beds) in Saudi Arabia; it offers comprehensive rehabilitation services, and it receives referrals from all over the country. Children with CP are enrolled in the pediatric rehabilitation program.

### 2.3. Procedure

This study was carried out with the approval by the SBAHC institutional review board (IRB) with IRB number: 72-2022-IRB. Informed consent was obtained from the study participants. Data were collected through electronic medical record review using a predesigned case report form, which included information on age, gender, type of CP, and the Gross Motor Function Classification (GMFCS) score. The questionnaire had two sections. The first section included information about caregiver age, gender, relationship with child, marital status, level of education, residency, type of living, and information on family size and income. The second section included the questions pertinent to KAB.

### 2.4. Questionnaire

The content validation of the questionnaire was reviewed by experts in the field and included one psychologist, one social worker, and one physiatrist expert in pediatric rehabilitation. The questionnaire focused on knowledge about CP (12 questions), attitude toward CP (10 questions), and caregiver behavior (9 questions). The case report form was written in English and translated into an Arabic version as the majority of participants were native Arabic speakers. First, the translated version of the questionnaire was piloted among 40% of the sample. The comments from the participants were received and addressed. After that, the questionnaire was distributed for the first response. Two weeks later, the questionnaire was distributed to the same participants for a second response. The internal consistency and reliability of the questionnaire were tested using Cronbach’s alpha. The results confirmed accepted values of the three domains of the questionnaire (0.66 for knowledge, 0.75 for attitude, and 0.87 for behavior).

### 2.5. Statistical Analysis

The collected data were double-checked, refined, coded, and subsequently transferred for analysis. The outcomes were tabulated in frequencies, percentages, means, and standard deviations, or they were presented in charts. Responses to knowledge were scored as follows: yes = 3, I do not know = 2, and no = 1. Responses to attitude and behavior were categorized and scored as follows: strongly agree = 5, agree = 4, neutral = 3, disagree = 2, and strongly disagree = 1. This 5-point Likert scale was re-categorized into a 3-point Likert scale as follows: strongly agree and agree as one point, neutral as one point, and disagree and strongly disagree as one point. The overall score was then calculated and interpreted as follows: 1.00–1.66 = poor (for knowledge) or negative (for attitude and behavior), 1.67–2.33 = moderate (for knowledge) or neutral (for attitude and behavior), 2.34–3.00 = good (for knowledge) or positive (for attitude and behavior). The data were subjected to a normality test, which revealed a non-normal distribution (Shapiro–Wilk test; *p* < 0.001). Accordingly, the differences between the participants in relation to the study variables were tested using the Mann–Whitney U test or Kruskal–Wallis test, as applicable. The statistical package for social sciences for Windows (SPSS v25, IBM Corp, Armonk, NY, USA) was used. The significance level was set at a *p*-value of <0.05.

## 3. Results

A total of 216 caregivers (mean age of 36.76 ± 7.85 years, ranging from 19 to 64 years old) participated in this survey. Among them, 88.9% were female, 50.5% were ≤36 years old, 89.4% were married, 53.7% were highly educated, 75.1% were unemployed, and 89.2% lived in urban areas. More than half of the participants (57.7%) owned their homes, 51.2% lived in apartments, and 67.5% received ≤ 10,000 SR as their monthly income. About 82.9% were mothers of CP children, 53.2% had ≤ 5 persons living in the family, and 91.7% had one child with CP. Regarding the CP children, more than half of them (52.8%) were males, and 32.9% were the first child in the family, with a mean age of 6.91 ± 3.25 years, ranging from 1 to 15 years old (49% were ≤6 years old and 51% were >6 years old). Spastic quadriplegia was the most common type (46.3% of cases), which was followed by spastic diplegia (44.9% of cases). GMFSC type IV was the most dominant (47.7%), which was followed by type III (31.0%). More details are presented in Table 1.

Figure 1 shows the overall knowledge, attitude, and behavior of the participants toward CP. Generally, the participants recorded good values for all variables. The mean value for attitude was 2.67 ± 0.20, which was followed by behavior (2.49 ± 0.36) and knowledge (2.46 ± 0.25).

Table 2 presents the responses of the participants to the knowledge questions. Generally, most responded correctly to the knowledge questions (64.7% of responses were correct), except for Question #4 “Do you think Cerebral Palsy is curable?”, where only 9.3% (n = 20) knew that CP is not curable, and Question #11 “Do you know what to do if your child chokes or has convulsions?”, where only 35.2% knew how to manage this problem. Regarding attitude (Table 3), most participants reported positive attitudes for all statements (51.0% and 28.8% of responses were “strongly agree” and “agree”, respectively), except for the statement “More care is given to the child with CP at the expense of the other children, where only 31.3% agreed with this. For behavior (Table 3), almost all statements were positively reported (36.8% and 31.1% of responses were “strongly agree” and “agree”, respectively). Table 4 shows the differences in the overall scores of the knowledge, attitude, and behavior according to the study variables. For knowledge and attitude, no significant differences were found between the participants for all variables (*p* > 0.05). For behavior, only two significant differences were found between the participants in relation to age of the CP child (*p* = 0.027) and type of CP (*p* = 0.037). However, for the type of CP, the detailed post hoc pairwise comparison revealed no significant differences between the pairs (Appendix A). Further linear regression analysis was performed to explore the most significant determinants, which revealed that both factors [age of CP child (*p* = 0.036) and type of CP (*p* = 0.015)] were significant determinants for behavior.

## 4. Discussion

The caregiver participation in the care journey of children with CP is essential. Their knowledge, attitude, and behavior significantly impact the quality of life and support provided to children with CP, especially in the early stages of their life. Our study stands out for its wide geographic and demographic representation within the country, unlike previous local studies that have reported on KAB, which have mostly included patients from their respective regions [29,30,31].

One of the largest KAB studies on CP in Saudi Arabia involved 450 parents with children of <17 years of age [30]. This study did not specifically target parents of children with CP, and it was intended to learn more about the knowledge, attitude, and behavior of parents in general toward CP. It showed that most participants had an overall good knowledge of CP, but they did demonstrate insufficient knowledge of disease aspects like causes, disease course, and prognosis. Overall, the results show a positive attitude of parents toward letting their child play with a child that has CP, but it also demonstrated a negative attitude toward hiring a person with CP or marrying a patient with CP. As this study was carried out in Al Baha area, which is located in the southwest of the country, it may only demonstrate the social attributes of a particular region; however, given the lack of data on the KAB toward children with CP, the results provide a background to the perception of the general population toward CP in KSA. Keeping the findings of this study in the background, and comparing them with the results of our study (which also showed a good mean score of KAB toward children with CP), it may be perceived that the general population in KSA may already be well familiar with the various aspects of the care of children with CP. This idea cannot be generalized by any means, and it renders the need for further large-scale comparative studies on the KAB of parents with children without and with CP; however, the lack of local epidemiological data on CP stands out as a priority. Various reports have emphasized the need of a CP registry in Saudi Arabia [32,33,34,35,36,37]; however, objective measures in this regard are lacking. Mushta et al. recently published (February 2024) a proposal for the establishment of the Saudi Cerebral Palsy Register (SCPR), which would be a crucial project in investigating and addressing the health care needs of individuals with CP in the country [38].

A critical ethnographic approach was employed in a local study by Mohamed Madi S. et al., in which six mothers of children with CP were interviewed [31]. Three primary themes were identified that specifically influenced and affected their experiences: (a) culture and religion, (b) motherhood and disability, and (c) community stigma and discrimination. Though the study sample was small, the investigators endorsed the need for individualizing the care of children with CP by considering the belief systems and experiences of the mothers. Given the unique socio-cultural and religious attributes in the Arab world, the findings of this study provide an insight on the KAB of parents toward children with CP. Though our study did not dwell into the belief systems and experiences of the caregivers, the study by Mohamed Madi S. et al. [31] provides a background for the possible influencing factors that shape the KAB of caregivers in our study.

Our study revealed a notable demographic trend among the participants, which indicated that the majority of caregivers were well-educated and unemployed. Interestingly, this finding mirrors a pattern observed in another study conducted in Saudi Arabia [39]. Our study found that caregivers with higher levels of education demonstrated a higher knowledge level regarding CP. This emphasizes the need for education in empowering caregivers to effectively support their children. Our findings also indicated that there were no significant differences in the knowledge, behaviors, and attitudes of caregivers related to the gender of children with CP. This finding is consistent with the results of a previous study [28].

This study revealed a prevalent misconception among participants regarding the curability of CP in addition to a lack of knowledge regarding how to handle their children in the case of choking or an episode of seizures. We found that more than half of caregivers did not know how to handle their children in the case of choking or when undergoing an episode of seizures. These are considered to be emergency situations. It is important to routinely include the education and training of caregivers on emergency situations. There are various guidelines and behavioral strategies to improve feeding techniques, prevent respiratory diseases, and to deal with emergency situations [40,41,42,43]. Improving the awareness and understanding of CP by using comprehensive educational strategies can enable caregivers to make individualized decisions and improve outcomes for children with CP.

This study revealed a significant difference in caregiver behavior based on the age of their child with CP. The caregivers of younger children demonstrated more positive behaviors compared to those who had older children with CP, thus indicating potential changes in parental responses over time. This suggests that parents may be initially experiencing heightened stress during the adjustment period to their child’s condition. With early family-centered rehabilitation interventions, children with CP show better functional outcomes [15]. Over time, parents adapt and develop coping strategies. The attitude of parents toward their children with CP is also crucial for their well-being. Parental positive attitude has shown a significant protective effect on parental distress [17]. On the other hand, overprotection attitudes have been reported to be prevalent among mothers of children and adolescents with CP [27]. The caregivers of children with CP that have severe impairments have lower behavior scores than their peers. Also, the type of CP also had a significant association with behavior scores. This could possibly be explained by the previously reported data, which show that children with quadriplegic CP require significantly more effort and time from their caregivers when compared to other patients with CP [5].

The majority of our sample (76.5%) reported having an income equal to or less than SAR 10,000. Given the special needs associated with CP, including the requirement for assistive devices, transportation expenses, and frequent visits to healthcare institutions, many caregivers expressed that caring for their CP child is costly and requires more expenses compared to their other children. Similarly, the living status (owned or rented house) demonstrated a significant relation with attitude. This may be indirectly related to the financial impact and economic strain involved in the care of children with CP. This finding appears to be a global issue, reflecting the financial challenges faced by caregivers of children with CP worldwide [44,45].

This study also highlights how Saudi Arabian family systems are different from those in western societies. In the United States, approximately one in three children live in a single-parent family, with the majority of families (83%) headed by a mother [46]. This is in contrast to Arab social norms, as in our study, where the majority of the caregivers were females who were married. Since children in single-mother families typically have poorer outcomes, across a range of measures [47], it can be argued that the chances of an additional burden of disability care could be potentially less in Arab family systems given that the majority of the children are in the care of mothers who are married. On the other hand, there are particular sociocultural challenges for children with disabilities and their caregivers in Saudi Arabia. A study by Madi et al. explored the belief systems of mothers of children with cerebral palsy, in which the participating mothers were found to make references to factors like the evil eye and Jinns as theological explanations for having a child with CP [31]. Similarly, the mothers’ denial, their unrealistic expectations, and the concerns of community stigmatization and discrimination were found to be considerable challenges that require a development of strategies and guidelines pertinent to these sociocultural attributes.

### Limitations

This study has several limitations. Firstly, the use of a self-reported questionnaire raises concerns about the possibility of invalid answers, as subjects may give a socially acceptable answer rather than an accurate one, especially for sensitive questions. Additionally, there is a risk of response bias, where participants may consistently answer with “yes” or “no” without fully considering the question’s meaning, which may impact data reliability. The lack of an objective parameter to define good knowledge, attitude, and behavior poses a challenge in standardizing the measurement.

## 5. Conclusions

Overall, the caregivers demonstrated good mean scores for knowledge, attitude, and behavior toward the care of children with CP. Strategies with a special emphasis on improving the behaviors of caregivers for children with quadriplegia need to be adapted. Similarly, the living situations of families need to be taken into consideration given its significant association with the attitude of caregivers. A considerable lack of knowledge in handling emergency situations by caregivers signifies a gap in care, which could have potentially life-threatening consequences. This highlights the importance of enhancing education programs with multidisciplinary teams to educate caregivers early on in the diagnosis process. It also stresses the need for caregiver support to manage critical scenarios effectively and to implement educational initiatives promoting positive behaviors and attitudes toward patients. Additionally, our findings highlight the importance of enhancing communication between healthcare providers, and it also emphasizes the need for promptly referring children with CP to relevant experts for the earliest possible intervention.

## Figures and Tables

**Figure 1 healthcare-12-00982-f001:**
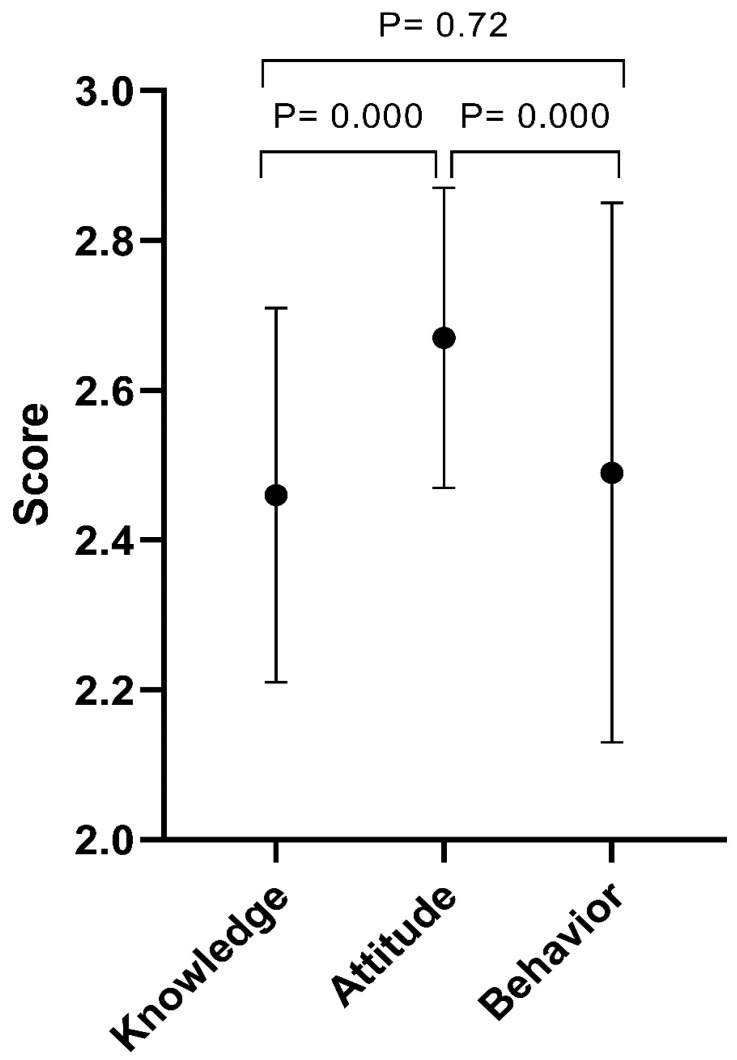
The knowledge, attitude, and behavior scores (mean ± SD) of the study sample. A Kruskal–Wallis test was used for the differences.

**Table 1 healthcare-12-00982-t001:** Characteristics of the study sample.

		Frequency	Percent
Gender of the child	Male	114	52.8
	Female	102	47.2
Birth order of the child	First child	71	32.9
	Subsequent child	145	67.1
Type of cerebral palsy	Dyskinetic	5	2.3
	Spastic diplegia	97	44.9
	Spastic hemiparesis	14	6.5
	Spastic quadriplegia	100	46.3
GMFCS classification	GMFSC II	12	5.6
	GMFSC III	67	31.0
	GMFSC IV	103	47.7
	GMFSC V	34	15.7
Gender of caregiver	Male	24	11.1
	Female	192	88.9
Age of caregiver	≤36 years	109	50.5
	>36 years	107	49.5
Relation to child	Father	29	13.4
	Mother	179	82.9
	Other	8	3.7
Marital status of the caregiver	Unmarried	23	10.6
	Married	193	89.4
Educational level	Illiterate	9	4.2
	Undergraduate	91	42.1
	Graduate	116	53.7
Living status	Owned	123	57.7
	Rented	90	42.3
Type of living	Apartment	109	51.2
	Villa	78	36.6
	Other	26	12.2
Working status	Unemployed	160	75.1
	Employed	53	24.9
Income	≤10,000 SR	141	67.5
	>10,000 SR	68	32.5
Residency	Urban	190	89.2
	Rural	23	10.8
Family size	≤5 persons	115	53.2
	>5 persons	101	46.8
Number of children with CP	One child	198	91.7
	More than one	18	8.3

CP = cerebral palsy and GMFCS = Gross Motor Function Classification System.

**Table 2 healthcare-12-00982-t002:** Responses to the knowledge questions.

	No	Do Not Know	Yes	Aver. Score
Do you know why your child is disabled?	37 (17.2)	13 (6.0)	165 (76.7)	2.58
Do you know what Cerebral Palsy disease is?	47 (21.8)	12 (5.6)	157 (72.7)	2.51
Do you think Cerebral Palsy is an inherited disease? *	17 (7.9)	44 (20.5)	154 (71.6)	2.63
Do you think Cerebral Palsy is curable? *	146 (67.9)	49 (22.8)	20 (9.3)	1.41
Do you think your child’s stiffness is related to Cerebral Palsy?	13 (6.1)	77 (36.0)	124 (57.9)	2.50
Do you think your child will have learning difficulties?	29 (13.6)	23 (10.7)	162 (75.7)	2.60
Do you think nutrition is important for your child’s growth?	11 (5.2)	15 (7.0)	187 (87.8)	2.79
Do you think saliva drooling and difficulty in swallowing are related to your child’s disease?	13 (6.0)	37 (17.1)	166 (76.9)	2.71
Do you think epilepsy is related to your child’s disease?	20 (9.3)	42 (19.5)	153 (71.2)	2.61
Do you think a rehabilitation program would improve your child’s quality of life?	2 (0.9)	10 (4.6)	204 (94.4)	2.94
Do you know what to do if your child chokes or has convulsions?	78 (36.1)	62 (28.7)	76 (35.2)	1.99
Do you think your child may have sensory, hearing, or visual defects?	36 (16.7)	78 (36.1)	102 (47.2)	2.31
Overall	17.4%	17.9%	64.7%	2.46

* Reverse-coded question; a higher number refers to a more correct response.

**Table 3 healthcare-12-00982-t003:** Responses to attitude and behavior statements.

	Strongly Disagree	Disagree	Neutral	Agree	Strongly Agree	Aver. Score
**Responses to attitude**						
Having a child with Cerebral Palsy disease gives a stigma. *	3 (1.4)	11 (5.1)	24 (11.1)	43 (19.9)	135 (62.5)	2.76
Parents have a role in the child’s disease. *	2 (0.9)	7 (3.3)	38 (17.7)	83 (38.6)	85 (39.5)	2.73
Parents should interfere when the child is aggressive toward himself or others.	5 (2.4)	10 (4.7)	11 (5.2)	70 (33.2)	115 (54.5)	2.74
More care is given to the child with CP at the expense of the other children.	46 (21.5)	70 (32.7)	31 (14.5)	47 (22.0)	20 (9.3)	1.75
Care of Cerebral Palsy patients is more costly.	10 (4.7)	11 (5.1)	8 (3.7)	63 (29.4)	122 (57.0)	2.74
The affected child is more dependent compared to his/her other siblings.	2 (0.9)	5 (2.3)	9 (4.2)	61 (28.2)	139 (64.4)	2.89
A sick child needs the parents’ support to cope with community needs.	2 (0.9)	4 (1.9)	3 (1.4)	41 (19.0)	166 (76.9)	2.93
A child with CP will have a normal life in the future.	3 (1.4)	18 (8.4)	30 (14.0)	89 (41.4)	75 (34.9)	2.65
A child with CP will be treated differently from other peers.	5 (2.3)	34 (15.7)	30 (13.9)	77 (35.6)	70 (32.4)	2.50
A child with CP will require seeking governmental or non-governmental support	0 (0.0)	0 (0.0)	2 (0.9)	45 (20.8)	169 (78.2)	2.99
Overall	3.6%	7.9%	8.7%	28.8%	51.0%	2.67
**Responses to behavior**						
You have a strong attachment to your sick child.	1 (0.5)	3 (1.4)	17 (7.9)	84 (38.9)	111 (51.4)	2.88
Your child changed your lifestyle.	4 (1.9)	11 (5.1)	31 (14.5)	85 (39.7)	83 (38.8)	2.69
More money is spent on the affected child.	4 (1.9)	20 (9.4)	25 (11.8)	69 (32.5)	94 (44.3)	2.61
Your relationship with the other children has been affected	17 (8.0)	60 (28.2)	51 (23.9)	53 (24.9)	32 (15.0)	2.01
Your child affected your social life.	21 (9.7)	51 (23.6)	45 (20.8)	61 (28.2)	38 (17.6)	2.13
Your child affected your work.	23 (10.8)	41 (19.2)	43 (20.2)	58 (27.2)	48 (22.5)	2.17
Looking after your child will be rewarded by God.	0 (0.0)	1 (0.5)	3 (1.4)	22 (10.3)	187 (87.8)	2.94
Having a sick child made others sympathize with you.	7 (3.3)	16 (7.4)	24 (11.2)	99 (46.0)	69 (32.1)	2.66
Having a child with CP disease has affected your psychological life.	25 (11.6)	30 (13.9)	44 (20.4)	69 (31.9)	48 (22.2)	2.29
Overall	5.3%	12.1%	14.7%	31.1%	36.8%	2.49

* Reverse-coded statement; a higher number refers to a more positive response.

**Table 4 healthcare-12-00982-t004:** Differences in the overall knowledge, attitude, and behavior scores according to the study variables.

		Knowledge		Attitude		Behavior	
		Mean ± SD	*p*	Mean ± SD	*p*	Mean ± SD	*p*
Gender of child	Male	2.44 ± 0.27	0.370	2.69 ± 0.19	0.153	2.52 ± 0.34	0.230
	Female	2.49 ± 0.23		2.65 ± 0.21		2.45 ± 0.38	
Age of child	≤6 years	2.46 ± 0.25	0.745	2.68 ± 0.20	0.190	2.54 ± 0.36	0.027
	>6 years	2.47 ± 0.26		2.65 ± 0.20		2.43 ± 0.35	
Birth order	First child	2.45 ± 0.26	0.476	2.64 ± 0.21	0.079	2.47 ± 0.38	0.655
	Subsequent child	2.47 ± 0.25		2.68 ± 0.20		2.49 ± 0.35	
Type of CP	Dyskinetic	2.52 ± 0.20	0.103	2.66 ± 0.27	0.756	2.73 ± 0.19	0.037
	Spastic diplegia	2.42 ± 0.27		2.68 ± 0.21		2.53 ± 0.36	
	Spastic hemiparesis	2.42 ± 0.32		2.71 ± 0.17		2.59 ± 0.32	
	Spastic quadriplegia	2.51 ± 0.23		2.65 ± 0.20		2.42 ± 0.36	
GMFCS classification	GMFSC II	2.42 ± 0.27	0.498	2.67 ± 0.13	0.303	2.59 ± 0.25	0.379
	GMFSC III	2.46 ± 0.26		2.68 ± 0.18		2.44 ± 0.38	
	GMFSC IV	2.48 ± 0.25		2.68 ± 0.21		2.51 ± 0.36	
	GMFSC V	2.41 ± 0.25		2.61 ± 0.23		2.45 ± 0.35	
Gender of caregiver	Male	2.41 ± 0.21	0.159	2.65 ± 0.21	0.508	2.58 ± 0.33	0.204
	Female	2.47 ± 0.26		2.67 ± 0.20		2.47 ± 0.36	
Age of caregiver	≤36 years	2.46 ± 0.25	0.939	2.66 ± 0.19	0.403	2.45 ± 0.36	0.101
	>36 years	2.46 ± 0.26		2.67 ± 0.21		2.52 ± 0.36	
Relation to child	Father	2.43 ± 0.22	0.539	2.66 ± 0.20	0.847	2.61 ± 0.32	0.151
	Mother	2.47 ± 0.26		2.67 ± 0.20		2.47 ± 0.36	
	Other	2.46 ± 0.25		2.69 ± 0.16		2.44 ± 0.38	
Marital status of caregiver	Unmarried	2.40 ± 0.28	0.246	2.64 ± 0.23	0.502	2.38 ± 0.33	0.141
	Married	2.47 ± 0.25		2.67 ± 0.20		2.50 ± 0.36	
Educational level	Illiterate	2.41 ± 0.35	0.888	2.68 ± 0.20	0.606	2.44 ± 0.40	0.929
	Low educated	2.45 ± 0.26		2.67 ± 0.22		2.49 ± 0.35	
	High educated	2.48 ± 0.25		2.67 ± 0.19		2.49 ± 0.36	
Living status	Owned	2.47 ± 0.24	0.607	2.64 ± 0.21	0.037	2.46 ± 0.36	0.300
	Rented	2.46 ± 0.26		2.70 ± 0.19		2.52 ± 0.36	
Type of living	Apartment	2.46 ± 0.26	0.617	2.69 ± 0.18	0.642	2.54 ± 0.35	0.095
	Villa	2.46 ± 0.24		2.66 ± 0.22		2.42 ± 0.37	
	Other	2.51 ± 0.23		2.62 ± 0.25		2.46 ± 0.38	
Working status	Unemployed	2.46 ± 0.26	0.952	2.67 ± 0.20	0.874	2.47 ± 0.36	0.369
	Employed	2.46 ± 0.24		2.66 ± 0.21		2.52 ± 0.36	
Income	≤10,000 SR	2.45 ± 0.27	0.117	2.67 ± 0.21	0.326	2.47 ± 0.37	0.344
	>10,000 SR	2.51 ± 0.23		2.67 ± 0.18		2.52 ± 0.36	
Residency	Urban	2.46 ± 0.25	0.479	2.67 ± 0.20	0.644	2.49 ± 0.37	0.679
	Rural	2.48 ± 0.30		2.67 ± 0.20		2.47 ± 0.32	
Family Size	≤5 persons	2.47 ± 0.24	0.967	2.68 ± 0.18	0.700	2.50 ± 0.36	0.575
	>5 persons	2.46 ± 0.27		2.66 ± 0.22		2.47 ± 0.36	
Number of children with CP	One child	2.46 ± 0.25	0.457	2.67 ± 0.20	0.931	2.49 ± 0.36	0.820
	More than one	2.51 ± 0.26		2.69 ± 0.18		2.48 ± 0.35	

CP = cerebral palsy and GMFCS = Gross Motor Function Classification System.

## Data Availability

The data presented in this study are available upon request from the corresponding author.

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
