# Peer review of "Caregiver Knowledge, Attitude, and Behavior toward Care of Children with Cerebral Palsy: A Saudi Arabian Perspective"

_healthcare, 2024, doi:10.3390/healthcare12100982_

Round 1

Reviewer 1 Report

Comments and Suggestions for Authors

In this study, the researchers assessed the knowledge, attitudes, and behaviors of caregivers towards children diagnosed with cerebral palsy, focusing specifically on a cross-sectional sample from Saudi Arabia. Here are my specific observations:

1. The authors effectively analyzed various factors influencing KAB, though it would be valuable to also examine caregivers' occupations and how stress within those professions affects KAB.

2. There appears to be a discrepancy between the text (lines 133-134) stating a higher percentage of female children compared to males, while Table 1 shows the opposite.

3. It would be helpful to mention the proportion of children under six years old versus older ones.

4. The statistical analysis method used for Figure 1 should be included in its legend.

5. While some variables showed similar percentages of participants, others such as caregivers' gender, marital status, employment status, residency, and number of children with CP displayed a notable imbalance. Therefore, the discussion section should be refined to address how these differences might impact the study outcomes.

Comments on the Quality of English Language

English language is fine. Lines 133-134 should be corrected.

Author Response

  1. The authors effectively analyzed various factors influencing KAB, though it would be valuable to also examine caregivers' occupations and how stress within those professions affects KAB.

Though the information on occupation of caregivers is interesting aspect to explore but this was not within the scope of our research. Most of the caregiver are housewives, which is culturally common in the region; however, on the same lines, in reference to education attributes of caregivers, high scores was noted among lower education level in knowledge and behavior.

  1. There appears to be a discrepancy between the text (lines 133-134) stating a higher percentage of female children compared to males, while Table 1 shows the opposite.

Thank you for your comment. The sentence has been corrected [lines 133-134].

  1. It would be helpful to mention the proportion of children under six years old versus older ones.

Thank you for your comment. The proportion has been added [line 135].

  1. The statistical analysis method used for Figure 1 should be included in its legend.

Thank you for your comment. The type of test (Kruskal Wallis test) was included in the legend of the figure [line 164].

  1. While some variables showed similar percentages of participants, others such as caregivers' gender, marital status, employment status, residency, and number of children with CP displayed a notable imbalance. Therefore, the discussion section should be refined to address how these differences might impact the study outcomes.

Reviewer comment related to findings pertinent to imbalance is explainable based on cultural attributes and social norms. For example, females are the most frequent caregivers who are mothers and mostly housewives, Children are born to married couples, so marital status findings are expected, as in our study. Our finding of families having more than two children with CP is similar to previously published literature.

Comments on the Quality of English Language

  1. English language is fine. Lines 133-134 should be corrected.

Thank you for your comment. The sentence has been corrected [lines 133-134].

Reviewer 2 Report

Comments and Suggestions for Authors

The current manuscript assessed the Caregivers’ Knowledge, Attitudes and Behaviors Towards Care  of Children with Cerebral Palsy in Saudi Arabia. The manuscript is well written and informative. Please find the following comments: 

- Line 39: Revise '' patients'' to '' children''

- Line 115-120: How the authors interpreted the sub-scores

- Line 120: Mention the name of the normality test

- Line 129: '' 89.4% were married'' does not make a sense. Revise

- The post-hoc analysis is not reported. Ho did you differentiate the significant differences of behaviour according to the type of CP (Table 4) 

Author Response

Line 39: Revise '' patients'' to '' children''

The second reviewer’s comment related to line 39 has been corrected.  

- Line 115-120: How the authors interpreted the sub-scores

Thank you for your comment. More details have been added as follows: “Responses to attitude and behavior were categorized and scored as follows: Strongly agree = 5, agree = 4, neutral = 3, disagree = 3, and strongly disagree = 1. This 5-point Likert scale was recategorized into a 3-point Likert scale as follows: strongly agree and agree as one point, neutral as one point, and disagree and strongly disagree as one point. The overall score was then calculated and interpreted as follows: 1.00–1.66 = Poor (for knowledge) or negative (for attitude and behavior), 1.67–2.33 = Moderate (for knowledge) or neutral (for attitude and behavior), 2.34–3.00 = Good (for knowledge) or positive (for attitude and behavior).” Lines [115-122].

- Line 120: Mention the name of the normality test

Thank you for your comment. The name of the normality test (Shapiro-Wilk test) has been added [lines 120-121].

- Line 129: '' 89.4% were married'' does not make a sense. Revise

The high rate of marital status among caregiver can be explained by the fact that divorce rate in our culture very low and marital status for housewife is high.

- The post-hoc analysis is not reported. Ho did you differentiate the significant differences of behaviour according to the type of CP (Table 4).

The post-hoc pairwise comparison revealed no significant differences between the pairs [Kindly refer to the Supplementary File] and attachment file  

Reviewer 3 Report

Comments and Suggestions for Authors

This study delves into the examination of the knowledge, attitudes, and behaviors (KAB) exhibited by caregivers in the provision of care for children diagnosed with Cerebral Palsy within the societal framework of Saudi Arabia. The investigation is conducted through a cross-sectional survey encompassing 216 participants, shedding light on notable discoveries pertaining to caregiver perceptions while also pinpointing areas of deficiency in knowledge and behavior. Of particular note are the inadequacies identified in emergency care, thus emphasizing the necessity for tailored educational interventions. This paper can be improved through:

1- expand on methodology, expand on sampling methods that used and justify the choice. 

2- enhance the discussion of findings and compare it to the existing literature, consider the sociocultural contexts and provide more valuable insights. 

3- address the potential biases to enhance the credibility of the data. 

4- provide recommendations for the healthcare practices and policy changes based on the identified gaps in the knowledge and behavior. 

Author Response

1- expand on methodology, expand on sampling methods that used and justify the choice. 

Thank you for your comment. Based on the nature of our study (specific patients with specific criteria), our population can’t be recruited randomly. Hence, we followed the convenience sampling to reach total study cohort . We have also specified the date or data collection [lines 82-85].

2- enhance the discussion of findings and compare it to the existing literature, consider the sociocultural contexts and provide more valuable insights.   

Comparisons have been made with previously published literature throughout the discussion. More information relevant to sociocultural attributes have been discussion with insights and recommendations. Lines 257-271. More references [46,47] have been added. 

3- address the potential biases to enhance the credibility of the data. 

Thank you for your comment. We have included a paragraph “Limitation” at the end of the Discussion section to explain the potential biases and limitations of the study [lines 272-279]

4- provide recommendations for the healthcare practices and policy changes based on the identified gaps in the knowledge and behavior. 

Based on this study it is recommended to empower educational program for caregivers of children with cerebral palsy. [269-271, and 287-290]

Reviewer 4 Report

Comments and Suggestions for Authors

This study entitled "Caregivers’ Knowledge, Attitudes and Behaviors Towards Care of Children with Cerebral Palsy; a Saudi Arabian Perspective." investigates the knowledge, attitudes, and behaviors (KAB) of parents with children with cerebral palsy (CP) in Saudi Arabia at Sultan Bin Abdulaziz Humanitarian City (SBHC). A cross-sectional survey of 216 caregivers was conducted, with 82.9% being mothers, half being ≤ 36 years old, 53.7% highly educated, and 89.2% living in urban areas. Spastic quadriplegia was the most common type. Results showed good values for KAB, but strategies to improve behaviors and family living situation need to be adapted. Even though the manuscript is primitive in analysis, the core content has value to consider for revision.   

Data should be subjected after descriptive statistics to deep statistical analysis to identify the significant association at the bivariate level using the regression analyses. The management of missing values shall be elaborated. 

Multivariate logistic regression analysis of primary variables on other variables shall be conducted to reveal the real association.

The bilingual questionnaire shall be added as supplementary data for the benefit of the readers.

Comments on the Quality of English Language

Moderate editing of English language required

Author Response

Thamk you very much for taking the time to review this manuscript. Please find the detailed responses below and the corresponding revisions/corrections highlighted/in track changes in the re-submitted files

should be subjected after descriptive statistics to deep statistical analysis to identify the significant association at the bivariate level using the regression analyses. The management of missing values shall be elaborated. 

Thank you for your comment. The regression analysis can be run once we found more than two significant results in the pairwise comparisons. As shown in Table 4, the pairwise comparison test revealed only significant differences in behavior in relation to age of CP child and type of CP. Accordingly, the regression analysis was run for this domain and the results have been reported [lines 158-165

-Multivariate logistic regression analysis of primary variables on other variables shall be conducted to reveal the real association.

Thank you for your comment. Kindly refer to our reply to the comment above this.

The bilingual questionnaire shall be added as supplementary data for the benefit of the readers

"The questionnaire used in the study is attached as a supplementary file 2 and   kindly  see the attachment 

Round 2

Reviewer 2 Report

Comments and Suggestions for Authors

The comparison according the type of CP was statistically significant (p=0.037). Therefore, there is a need for post-hoc analysis

Author Response

"Please see the attachment

Reviewer 4 Report

Comments and Suggestions for Authors

Revised MS can be accepted.

Comments on the Quality of English Language

Moderate editing of English language required

Author Response

Thank you for your feedback. The English version has been reviewed for corrections and improvements.